# Microstructural and Mechanical Properties of Novel Co-Free Maraging Steel M789 Prepared by Additive Manufacturing

**DOI:** 10.3390/ma15051734

**Published:** 2022-02-25

**Authors:** Zbigniew Brytan, Mariusz Król, Marcin Benedyk, Wojciech Pakieła, Tomasz Tański, Mengistu Jemberu Dagnaw, Przemysław Snopiński, Marek Pagáč, Adam Czech

**Affiliations:** 1Department of Engineering Materials and Biomaterials, Faculty of Mechanical Engineering, Silesian, University of Technology, 44-100 Gliwice, Poland; zbigniew.brytan@polsl.pl (Z.B.); tomasz.tanski@polsl.pl (T.T.); przemyslaw.snopinski@polsl.pl (P.S.); 2Paks’D Sp Zoo, Strzelecka 74, 43-100 Tychy, Poland; mbenedyk@paksd.co; 3Department of Mechanical Engineering, Institute of Technology, Wollega University, Nekemte P.O. Box 395, Ethiopia; mengistuj@wollegauniversity.edu.et; 4Center of 3D Printing Protolab, Department of Machining, Assembly, and Engineering Technology, Faculty of Mechanical Engineering, Technical University of Ostrava, 17 Listopadu 2172/15, Poruba, 708 00 Ostrava, Czech Republic; marek.pagac@vsb.cz; 5Department of Lightweight Structures and Polymer Technology, Chemnitz University of Technology, 09111 Chemnitz, Germany; adam.czech@mb.tu-chemnitz.de

**Keywords:** SLM, LPBF, M789 steel, oxide inclusions, heat treatment, microstructure, mechanical properties

## Abstract

This research aims to characterize and examine the microstructure and mechanical properties of the newly developed M789 steel, applied in additive manufacturing. The data presented herein will bring about a broader understanding of the processing–microstructure–property–performance relationships in this material based on its chemical composition and heat treatment. Samples were printed using the laser powder bed fusion (LPBF) process and then the solution was annealed at 1000 °C for 1 h, followed by aging at 500 °C for soaking times of 3, 6 and 9 h. The AM components showed a relative density of 99.1%, which arose from processing with the following parameters: laser power of 200 W, laser speed of 340 mm/s, and hatch distance of 120 µm. Optical and electron microscopy observations revealed microstructural defects, typical for LPBF processes, like voids appearing between the melted pools of different sizes with round or creviced geometries, nonmelted powder particle formation inside such cavities, and small spherical porosity that was preferentially located between the molten pools. In addition, in heat-treated conditions, AM maraging steel has combined oxide inclusions of Ti and Al (TiO_2_:Al_2_O_3_) that reside along the grain boundaries and secondary porosities; these may act as preferential zones for crack initiation and may increase the brittleness of the AM steel under aged conditions. Consequently, the elongation of the AM alloy was low (<3%) for both annealed and aged solution conditions. The tensile strength of AM M789 increased from 968 MPa (solution annealed) to 1500–1600 MPa after the aging process due to precipitation within the intermetallic η-phase. A tensile strength and yield point of 1607 ± 26 and 1617 ± 45 MPa were obtained, respectively, after a full heat treatment at 500 °C/6 h. The results show that 3 h aging of solution annealed AM M789 steel achieves satisfactory material properties in industrial practice. Extending the aging time of printed parts to 6 h yields slightly improved properties but may not be worth the effort, while long-term aging (9 h) was shown to even reduce quality.

## 1. Introduction

Additive manufacturing (AM), or 3D printing, has revolutionized the manufacturing world through its rapid and geometrically complex capabilities, alongside its economic benefits. Over the past decade, countless businesses in the automotive, energy, aerospace, medical, and even food industries have adopted this approach [1,2]. Additive manufacturing processes involve the building of three-dimensional parts by adding thin layers of material guided by a digital CAD model. This feature allows the production of complex or customized parts directly from a digital model, thereby excluding the need for conventional tools such as dies, forms and casting molds; it also decreases production expenses due to the reduction in the number of necessary manufacturing steps. Additionally, AM allows for on-demand production of parts alongside a reduction in the part count because of the lower number of assembling constituents in the design. These factors explain why AM has attained extensive attention over the past few years, especially in the industrial sectors described above [1,2,3,4,5].

Today, AM has reached a critical acceptance level, as evidenced by the rapid growth of commercial AM systems due to concurrent advances in the development of cost-effective industrial lasers, inexpensive high-performance computing hardware and software, and technological progress in the production of metal powder feedstocks [1,2,3]. Among the various AM techniques, the selective laser melting (SLM) process, also known as laser powder bed fusion (LPBF), uses a high-energy laser as a heat source that selectively melts a predeposited powder bed; it is currently considered one of the most advanced and promising AM techniques [4,5,6]. This process, whose input material is a metal powder, is characterized by a number of key parameters, such as laser beam size and output power, scanning speed, and layer thickness [7]. However, despite the benefits of AM, one major limitation is its creation of unfeasibly high crack sensitivity in known commercial alloys. For this reason, new alloys must be designed and developed to maximize the benefits of this emerging technology [8]. The material most frequently used, in addition to the austenitic alloys AISI 316L or AISI 304, is 18% Ni maraging steel X3NiCoMoTi18-9-5 (EN: 1.2709, known also as 18Ni(300), BÖHLER W722 AMPO, Maraging 300) that is hardened by nanometer-sized intermetallic precipitates. The well-balanced property relationship between hardness, strength, toughness, and ductility, separate from the easy LPBF processability, leads to this alloy being one of the most LPBF-produced steel powders. However, the microstructure of X3NiCoMoTi18-9-5 does not exhibit corrosion resistance, due to the lack of chromium that forms a surface-protective passive layer. If corrosion resistance is required for specific applications, then AM designers and engineers must change their material selection to corrosion-resistant austenitic or precipitation hardened maraging steels such as X5CrNiCuNb17-4-4 (EN: 1.4548, BÖHLER N700 AMPO, 17-4 PH). However, these steels exhibit both lower strength and hardness compared to X3NiCoMoTi18-9-5. For such reasons, a modified Co-free maraging steel grade (M789) has been developed, in which chromium and nickel are responsible for the formation of nickel martensite that increases corrosion resistance due to the presence of chromium, while titanium and aluminum form precipitates that strengthen the microstructure during the aging stage of heat treatment.

Data available in the literature on the 3D printing of M789 maraging steel powder are very limited. The M789 powder was launched within the past 3 years, and detailed research in its field is still in progress. The first work was carried out by Turk [9] and then by Pallad et al. [8,10] and is still being carried out [11]. Pallad et al. [8,10] reported on M789 steel; the hardness increases from about 31 HRC to around 52 HRC and tensile strength from about 1019 MPa to around 1798 MPa after heat treatment, i.e., at 500 °C for 120 min. This observation was also stated by Turk et al. [9] on M789 steel; the maximum hardness of 52 HRC and tensile strength around 1820 MPa was obtained by heat treatment in 500 °C for 180 min. Most of the available research focuses on maraging steel with the addition of Co [12,13], while Co-free grades are a novelty. Post-processing heat treatment of AM maraging steel is an important issue in the optimisation of the final mechanical properties of final parts. According to the manufacturer’s recommendations, the maximum effect of precipitation strengthening for Co-free M789 steel can be obtained at a temperature of about 500 °C for 3 h [14]. However, for classic grade 1.2709 with Co addition, ageing treatment of 3–6 h [15,16] and even 10 h is recommended [17].

In this work, a newly designed iron-based alloy based on grade 250 maraging steel, commercially known as M789 steel, is characterized. M789 steel combines the printability of maraging steel with an improved corrosion resistance. The excellent printability of M789 steel can be attributed to the absence of brittle intermetallics and the low amount of carbides formed after solidification. This alloy provides a well-balanced property relationship between hardness, strength, toughness, ductility, and corrosion resistance to combine the characteristic properties of maraging steel and stainless steel, such as X3NiCoMoTi18-9-5 and X5CrNiCuNb17-4-4 [9].

The aim of this research is therefore to check the effect of heat treatment of elements printed with SLM technology on the AM125 device manufactured by Renishaw using the commercially available M789 maraging steel and to evaluate the aging time (3, 6, and 9 h) after solution annealing on the microstructure and mechanical properties obtained.

## 2. Materials and Methods

In the experiment, M789 steel powder (0.02% C, 12.2% Cr, 10% Ni, 1% Mo, 0.06% Al, 1% Ti, Fe), manufactured by Voestalpine BÖHLER Edelstahl GmbH & Co KG, was used to print basic samples in the LPBF process. This is a martensitic maraging steel, which contains a very low percentage of carbon and alloyed with chromium, nickel, molybdenum, aluminium, titanium, and other negligible elements; the entire nominal chemical composition specification of the powder is shown in Table 1. The M789 powder is a gas atomized powder with particle diameters that range between 15–45 µm. Before the printing, an analysis of the powder was performed in relation to its morphology and particle distribution. The morphology of the powder is an important characteristic that affects the deposition of the metal powder (by the wiper) in terms of flowability and packing density. The powder morphology was evaluated using a scanning electron microscopy (SEM) Supra 35 from Zeiss Company. The particle size distribution was determined by a laser diffraction technique that uses the Fraunhofer model of light scattering by particles. The analysis was executed using wet dispersion via an Analysette 22 MicroTec apparatus from Fritsch GmbH.

For the metallographic examination, a LEICA MEF4A light optical microscope (LOM) with a Leica Image Analyzer was used on the as-built and heat-treated components. The metallographic specimens were fabricated using a conventional procedure that consists of grinding, emery paper polishing, and then cloth polishing. SEM observations were made on the electrolytically etched metallographic samples within 10% oxalic acid, and 3–6 volts were applied for 5–60 s. The density of the manufactured components was estimated by an image analysis ImageJ of the unetched samples, which enabled a measuring of the percentage area of porosity on the polished surfaces. The average values and standard deviations were estimated based on observations in five different regions.

The AM125 RENISHAW system was used to fabricate the components using the LPBF technique. This scheme is characterized by a Ytterbium (Yb) fiber laser with a maximum laser power of 200 W, a scan speed of 2000 mm/s, and a wavelength of 1074 nm. The designed components were manufactured on a mild steel platform within an Ar inert gas atmosphere at an oxygen level that is below 10 ppm. The preheating of the substrate is not required for this type of material. A meander scanning strategy was used following a rotation of 67° after every layer is laid. Currently, many works have been performed that relate the influence of the manufacturing parameters of SLM technique for materials composed of M789 steel [9,10,11]. The energy density, *E*_d_, is a critical parameter in the selective laser melting technique. It correlates with the laser power, *P*, scan speed, *V*, hatch distance, *h*, and layer thickness, *t*, in the presented work through the following equation [18]:(1)$$E_{d}=\frac{P}{Vht\;}\;\left(\frac{J}{{\mathrm{mm}}^{3}}\right)$$

To manufacture almost fully dense components, an optimization was applied to estimate the processing parameters. Using the different process conditions listed in Table 2, the powder beds were selectively fused layer-by-layer until the final 3D component was completed.

The heat treatment of the LPBF samples was performed as follows: solution annealing at a temperature of 1000 °C, 1 h soaking time, and air cooling to room temperature. Next, using the conditions that aging is set at 500°C and the heating rate is 10 °C/min, an isothermal holding in Ar atmosphere was provided for 3, 6, and 9 h, then cooled within the furnace to ambient temperature. The post-heat treatment conditions were selected based on values found in the literature [10,19,20], in which the manufacturer describes aging at 500 °C as the ideal temperature for producing the optimum properties of printed M789 steel. The soaking time at the aging temperature was varied, which enabled an evaluation of its effects on the microstructural and mechanical properties of the material. The aging treatment was accomplished within a high-temperature HT-2100 G-Vac Graphite-Special vacuum furnace from Linn High Therm GmbH.

X-ray diffraction (XRD) patterns were collected using an X-Pert PRO instrument. For the X-ray diffraction analysis, a Co target and a scan rate of 0.01 step/s and a scan range for 2*θ* between 30 to 110° were used. The X’Pert HighScore Plus was used for phase identification and quantitative analysis. Retained austenite content was calculated with the RIR (Reference Intensity Ratio) method. The microstrain and dislocation density were calculated for the martensite phase peaks. The full-width at half-maximum (FWHM, *β*) was estimated by using a profile fitting. The crystallite size *D* (in units of nanometers) was then calculated with the Scherer equation (*D* = *kλ*/*β*co*sθ*), the dislocation density *δ* (nm^−2^) from *δ* = 1/*D*^2^ and the microstrain *ε* from *ε* = *β*/4tan*θ*. The tensile test was performed using a Zwick Z100 tensile test machine under the PN-EN ISO 6892-1 standard. Tensile test samples were prepared using the ISO-2740 standard for the “dogbone” shape samples. The Charpy impact test was performed on V-notch samples with PN-EN ISO 148-2. During the printing process, the samples for the tensile test and the Charpy impact test were oriented in a horizontal direction (i.e., the horizontal samples), which is perpendicular to the printing direction. The hardness values were measured using a Zwick ZHR 4150 TK hardness tester, in HRC scale, according to the ISO 6508 standard, at the surface under printed conditions and at the cross section within the core of the material.

## 3. Results and Discussion

The evaluation of the heat treatment parameters (aging for 3, 6, and 9 h) on the properties of the AM M789 maraging steel began with the base powder examination presented in Section 3.1. Next, the effect of aging on the microstructure of the printed samples was studied (Section 3.2). The mechanical properties derived from the tensile test, the hardness measurements in the cross section and on the surface, as well as in the Charpy impact test, are discussed in Section 3.2. The research is then summarized and conclusions drawn.

### 3.1. Precursor Powder and As-Printed Sample Characteristics

The shape of the particle size distribution curve is relatively narrow and close to symmetric for the maraging powder M789. The median diameter of D50 is 29.0 µm and the diameter range of D10-D90 is between 15.7–48.3 µm (Figure 1). Figure 2 represents a typical spherical morphology for the gas atomized M789 powder. Evaluation of the powder morphology revealed that some of the particles are not fully spherical or include satellite particles. The powder batch with particles’ near spherical morphology is characterized by slightly reduced flowability and hence low apparent density. As a result, the SLM process leads to the famous “balling” effect, and high porosity can be observed in the samples.

An optical microscope was used to assess the melting pool and grain structures under as-printed conditions (Figure 3a). Overlapping melt pools were observed in the fusion line region typical of the AM process. Interesting results were shown by the linear EDS analysis at the border of the melted pool (Figure 3b). An increase in the share of aluminum and titanium is visible and, on the line of analysis 3, a precipitation enriched with both elements is revealed. However, its morphology does not differ from that of the surrounding steel microstructure.

Figure 4 illustrates, using light optical microscopy (LOM), the microstructural defects in the horizontal cross sections of the samples in the form of keyholes, caves, and gas pores; this is typical for materials created using AM technology. The porosity measured by image analysis under different manufacturing conditions is presented in Figure 5. The highest relative density value was measured at 99.1 ± 0.4% and corresponds to the fifth set of printing parameters in Table 2—*P* = 200 W, *v* = 340 mm/s and *h*_d_ = 0.12 mm. It is noteworthy that numerous defects arise in components that are printed at higher laser speeds.

### 3.2. Effect of Aging on the Microstructure of the Printed Samples

The martensitic structure in this study was body-centered cubic (bcc) due to the low carbon concentration in the M789 maraging steel [21]. Therefore, XRD patterns were interpreted based on the bcc structure, and a good graph fitting for the pattern was confirmed. The X-ray diffraction analysis of M789 steel revealed strong peaks derived from the martensitic phase α’ and weak austenite γ peaks (Figure 6). The diffraction peaks of the retained austenite Fe-γ (111), (200), (220), (311) and martensite Fe-α’ (110), (200), (211), (220) were clearly identified from the X-ray patterns for all studied heat treatment conditions. The retained austenite content of 6% in as-solution annealed conditions was slightly increased by 8% in the aging conditions, regardless of soaking time, when studied over the range 3–9 h (Table 3). The highest intensity peak for martensite Fe-α’ (110) in solution annealed conditions at 52.1212 2Θ° shows a strong shift with the aging of time. The Fe-α’ (110) peak is maximally shifted to 52.19379 2Θ° in 3 h aged steel. Moreover, after 6 h and then 9 h of aging, the peak location of Fe-α’ (110) is shifted towards the lower 2Θ° values, which represents a return to values that are close to the solution-annealed conditions (Figure 7).

The full width at half maximum (FWHM, *β*) of the peaks corresponding to martensite increased after aging (Table 3). The FWHM of the diffraction peaks may be related to various material properties, such as grain distortion, dislocation density, and residual stresses [22]. The increase in FWHM and the widening of the X-ray peak were associated with an increase in the stacking faults and structural disorder, alongside the presence of tensile stress in the material, while a relaxation of the tensile stress decreased the FWHM [23]. The linear increase in the FWHM of the XRD peak was also related to increases in the hardness and density of the point defects that alter the crystallinity and grain boundary mobility [24]. Peak broadening of martensite is commonly associated with a high amount of lattice defects; for example, the peak shape can be used to predict the dislocation density in martensite. When analyzing the XRD parameters of the martensite peaks (Table 3), it was apparent that the number of lattice defects, the calculated dislocation density, *δ*, and microstrains, *ε*, were connected. Thus, the residual stresses (the increase in FWHM) were related to the precipitation of the secondary phases that resulted in lattice distortions in samples that were subjected to prolonged aging, while the crystallite size, *D*, decreased with aging time.

As confirmed by the XRD analysis, the microstructure of the solution-annealed maraging M789 steel was composed of a martensitic matrix and some retained austenite (Figure 8a). Aged M789 steel shows a martensitic matrix with retained austenite on the grain boundaries and nanometer-sized round precipitates inside the grains and along the grain boundaries. The martensitic needle-like structure of the heat-treated steel is comparable to the microstructure of conventionally fabricated maraging steel.

Additionally, numerous microstructural defects that typically form during the laser powder bed fusion (LPBF) additive manufacturing process were identified in the steel microstructure, i.e., voids are observed between the melted pools of different sizes with round or crevice geometry (Figure 8b). This means that unmelted powder particles can reside inside such cavities, and small spherical porosities can form that preferentially locate between the molten pools.

The morphology of the nanoscale precipitates that are observed in the maraging steel matrix is similar to those described in the literature [8,10,19]; they can be described as an intermetallic compound with the general formula ETA-Ni_3_(Al,Ti). In the case of steel M789, the addition of Ti enables the formation of Ni_3_Ti precipitates. On substituting the remaining Ti in the matrix by Al, Ni_3_Al precipitates form during the aging stage of the heat treatment. The study presented here did not analyze in detail, the precipitation of the secondary phases such as ETA-Ni_3_ (Al, Ti) (*η*-phase), which are responsible for the basic mechanism that strengthens the maraging steel. Depending on the alloy composition and heat treatment conditions, the strengthening mechanism during the ageing heat treatment relates to the precipitation of various intermetallic phases, such as Fe_2_Mo, Fe_7_Mo_6_, Ni_3_Ti and NiAl. Depending on the intermetallic composition, the effect on the age hardening of the nickel-rich martensite of the maraging steels can be strong (due to the addition of Ti or Be), moderate (when alloyed with Al, Nb, Mn, Si, Ta or V) or weak (Co, Cu or Zr) [25].

A closer examination of the grain boundaries revealed precipitates residing along some of the melting track boundaries (Figure 9). Precipitates are preferentially located within regions of grain boundary concentrations and zones of multiple solidification during the LPBF process. Precipitations are also present in the areas of porosity and within cracks between grain boundaries. The presence of secondary phases may enable preferential sites for crack formation, while diffusion processes that occur during aging may favor the formation of secondary porosity during the precipitation of massive secondary phases at the grain boundaries. The shape of the secondary phase is round, spherical, or lenticular (Figure 9), and it is anchored at the grain boundary and grows along it (Figure 8b). Massive precipitates of oxides (oxygen content between 20–30%) are preferentially composed of Al and Ti; they consist of approximately 25–39% Al, 7–10% Ti, 3–6% Cr, 1–4% Ni and 0.1–0.4% Mo (Figure 10, Table 4). The sizes of the oxides (i.e., TiO_2_:Al_2_O_3_) are less than 30 µm with a longer border or less than 10 µm. The appearance of these combined oxide inclusions of Ti and Al was also confirmed in the additively manufactured maraging steel (1.2709, 18Ni (300)) [11,12]; their presence relates to a decrease in the plastic properties of the maraging steel under aged conditions.

The phenomenon of surface oxide formation during additive manufacturing of maraging steel is still under intensive study; results in the literature show that, in the AM of steel that contains alloying elements with a high potential for oxidation, an oxide layer containing Al and Ti will be created on top of each layer. During the subsequent stages of printing, the oxide layers that are formed will be destroyed and mixed with liquid metal as a consequence of Marangoni flow. As a result, there is an accumulation of oxides at the periphery of the melt tracks of the bulk material, which form repeated pattern strips and massive, irregular-shaped oxide inclusions (which are mostly round, crescent or lenticular (Figure 8 and Figure 9) [18,26].

### 3.3. Mechanical Properties

The mechanical properties of the maraging steel were evaluated using five measurements and three properties were analyzed closely, since their registered values have a difference that is less than 3%. The average ductility of the heat-treated samples (*A*_5_ = 2%) is lower than the solution annealed sample (*A*_5_ = 3.5%). Similarly, the toughness value of the solution annealed specimen (i.e., 20 J) decreased to 10 J after the heat treatment with aging. On the other hand, the tensile strength *R*_m_ = 968 MPa, and the yield strength *R*_p0,2_ = 869 MPa, in solution annealing conditions are considerably lower than those of the heat-treated samples, which reach *R*_m_ = 1550–1615 MPa and *R*_p0,2_ = 1520–1607 MPa. The highest mechanical properties were obtained for 3 h and 6 h soak treatments at 500 °C, while a prolonged aging for 9 h resulted in a slight decrease in the yield, *R*_p0,2_, and ultimate tensile strength, *R*_m_.

The mean superficial hardness of the AM M789 steel, measured on the polished surface, increased from 26 HRC under solution annealing conditions to a range between 42–46 HRC after aging. When 3 h of aging was applied at 500 °C the result was 46 HRC, while the longest time of heat treatment resulted in slightly lower values, such as 42 HRC. The core hardness (measured on the cross section) of the solution annealed surface showed a higher fluctuation (with a mean 41 HRC) than in the case of the heat-treated samples, for which the surface showed a uniform hardness distribution (52–53 HRC) with few scattered values (Figure 11). The fluctuation in the hardness was linked to the subsequent laser fusion line and corresponded to periodical zones inside which structure defects accumulate. The mechanical properties of the AM M789 steel subjected to aging heat treatment are summarized in Table 5.

The fractured surface of the Charpy samples shows a mixed type of ductile and brittle fracture and, at higher magnification, numerous fracture mechanisms were detected. The solution annealed fracture surface is covered by transgranular cleavage zones close to the porosities and voids between the melted pools, where no fully melted powder particles can be seen (Figure 12a). The brittle cleavage zones are uniformly distributed on the fracture surface and they mostly contain a round and oval shape (Figure 12b). Therefore, it can be presumed that these zones occur at the positions of microstructural defects, which is typical for LPBF technology, such as voids between molten pools and the borders of molten pools. The presence of voids with powder particles entrapped within them, and the balling effects related to an insufficient wetting ability of the substrate layer by the molten material, causes liquid spheroidizing in preferential places for stress concentration and a reduction in the material plasticity (Figure 12c). Insufficient melting is seen in the nonoverlapped regions between adjacent melt pools, thus void formation becomes typical for the AM process and it is difficult to completely eliminate (however, a significant reduction is possible via an optimization of the processing parameters). In addition, the entire surface has the characteristics of a ductile fracture surface with smaller micro-dimples and has larger plastic flow zones that result from slip deformation.

The surface fractures of the aged specimens were similar in nature for each case, i.e., there were small longitudinal gaps oriented in one direction (Figure 13c) and large voids with balling particles present (Figure 13b). However, the size of these defects was greater than in the case of the solution-annealed condition. A closer examination of the fracture surface provided evidence of a transgranular cleavage fracture mode that was composed of fine planes, which were oriented in different directions and devoid of large areas with a uniform flat fracture (Figure 13a).

It should be emphasized that a commercial powder was used, without any preliminary preparation applied to it, alongside a subsequent baseline optimization of the printing parameters. The relative density was obtained at the level of 99%, which is a good prognosis for the possible properties of the AM steel, but despite this, the ductility of the AM samples was quite low. The results of mechanical properties (i.e., yield point and tensile strength of 1500–1600 MPa) are below the maximum values available in the literature (i.e., 1800 MPa) [8,10,19]; these results are probably associated with the presence of the combined oxide inclusions of Ti and Al (TiO_2_:Al_2_O_3_) in the steel microstructure, which were coagulated and concentrated in the area of the grain boundaries during the sintering process. Consequently, they contribute to the reduction in the plastic properties of the AM steel, which appear in the samples under solution annealed conditions. Similar oxide inclusions, described previously in other works [11,12], were also associated with a reduction in plastic properties. Certainly, for steel powders intended for additive manufacturing, the surface conditions are essential. This includes the morphology of the powders, the uniformity of the particle shapes, and the oxidation levels of the metallic powder (it is noteworthy that the presence of crushed powder particles may enhance oxidation). In the analyzed case of M789 maraging steel, the influence of the shielding gas during sintering should be reanalyzed. In addition, further research is required to determine the oxygen content in the steel powder and its level during the AM process. During laser processing, oxidation of the processed powder is one of the key concerns in AM [27]. This is because the high amount of oxygen in the powder may influence the melt pool dynamics (i.e., by redirecting the melt flow, changing the geometry of the solidified track, etc.), the powder wettability, and the laser absorptivity; thus, initiating oxide formation on the powder particles surface.

Data available in the literature on the 3D printing of M789 maraging steel powder are very limited. The M789 powder was launched within the past 3 years, and detailed research in its field is still in progress. The first work was carried out by Turk [9] and then by Pallad et al. [8,10,19] and is still being carried out [11]. The works in this area come from practically one international research group. As mentioned above, the mechanical properties of the printed parts in this study are below the maximum values available in the literature where well-optimised process conditions are used. Certainly, research in this area by various research centres may provide a more complete characterization of the tested steel, taking into account various devices and printing strategies. On this basis, it is evident that the printing conditions and their optimization for a given system, device, and printing strategy are extremely important. Manufacturers usually provide exact printing parameters along with the type of device for which they were obtained. Therefore, in each case, they should be adapted to the device and optimized for the final shape of the element. From a practical point of view, it can be concluded that it will be possible to successfully obtain properties at the level of 90% of the declared mechanical properties. However, approaching the maximum values will require careful optimization.

In addition, the heat treatment for AM 789 steel recommended by manufacturers and powder suppliers [14] recommends aging after solution annealing for 3 or 6 h. Such conditions were analysed in the study together with prolonged aging to 9 h. As a result of this work, it was revealed that 3 h aging of AM M789 steel allows satisfactory material properties in industrial practice. Extending the aging time of printed parts to 6 h gives slightly but higher properties and may not be worth of effort, while long-term aging (9 h) even lowers them.

The method of storing powder materials can also be important. In this work, the powder was stored in a standard manufacturer container for at least one year from the date of purchase and more from the day of production. The tendency to oxidize, the proportion of powder particles that are not perfectly round, and some of the powder particles with damaged surface may influence the tendency for the formation of oxide inclusions. Such oxides have been disclosed in the work and are associated with the deterioration of the mechanical and plastic properties of the printed samples. The above factors should be taken into account during the commercial application of the test powder in industrial practice.

In industrial practice, the results obtained may also be influenced by the interchangeability of the use of different powders in a given machine. Although the powder container and the sintering chamber were cleaned after the powder was changed, this influence cannot be ignored. In this study, every effort was made to obtain the highest purity of the LPBF process, and the powder used was stored in the classical way without additional safeguards.

## 4. Conclusions

The effect of heat treatment (i.e., solution annealing with variable soaking times at a temperature of 500 °C) on the microstructural and mechanical properties of a novel Co-free maraging steel M789 was investigated. From the presented results, the following conclusions can be drawn:
The analysis revealed significant differences in the results for the relative density, which arises from the high pore depths and voids between the scan lines. The highest relative density was close to 99.1%, which was found for components fabricated with the parameters: laser power = 200 W, laser speed = 340 mm/s, and hatch distance = 120 µm.The microstructure of the AM M789 maraging steel is composed of a martensitic microstructure with retained austenite. The austenite content under solution annealing was 6% and it was slightly increased to 8% after the aging stage 3–9 h.The XRD parameters of the martensitic peaks portray the amount of lattice defects, dislocation density, *δ*, and microstrains, *ε*. While the residual stresses (i.e., the increase in FWHM) relates to the precipitation of secondary phases that results in lattice distortions occurring in samples subjected to prolonged aging. In addition, the crystallite size, *D*, decreases with aging time.The maraging steel under heat treated conditions, regardless of the time of aging, includes the presence of combined oxide inclusions of Ti and Al (TiO_2_:Al_2_O_3_) along the grain boundaries and secondary porosity. These are considered metallurgical defects that can act as preferential zones for the initiation of cracks and may increase the brittleness of the AM steel under aged conditions. The effects of such oxide inclusions are visible in the form of relatively low plastic properties (elongation after fracture, *A*_5_ < 2%) of the printed parts, both under solution annealed and annealed and aged conditions.The mechanical properties of the printed parts in this study, in terms of tensile and yield strengths (i.e., ~1600 MPa), are lower by ~11% than those values found in the literature (i.e., ~1800 MPa).When comparing the heat-treated AM M789 maraging steel, the samples under solution annealed conditions have the lowest hardness and tensile strengths. However, subsequent ageing treatment causes improvements to the mechanical properties, i.e., tensile and yield strengths, and decreases the already low levels of elongation and toughness. The optimal material properties were obtained when the ageing step lasted for 6 h at 500 °C, which provided a core hardness of 53 HRC, a superficial hardness of 42 HRC, and yield and tensile strengths as high as 1617 and 1602 MPa, respectively. The difference between the yield and tensile strength was low (i.e., 10–30 MPa), which demonstrates the low ductility of the AM maraging steel. Further prolongation to 9 h for the heat treatment ageing step causes a subsequent decrease in the mechanical properties (approximately by 50 MPa for the yield and tensile strengths).This work revealed that the steel powder characteristics play an important role in achieving the final properties of the M789 maraging steel, especially in terms of the plastic properties.

## Figures and Tables

**Figure 1 materials-15-01734-f001:**
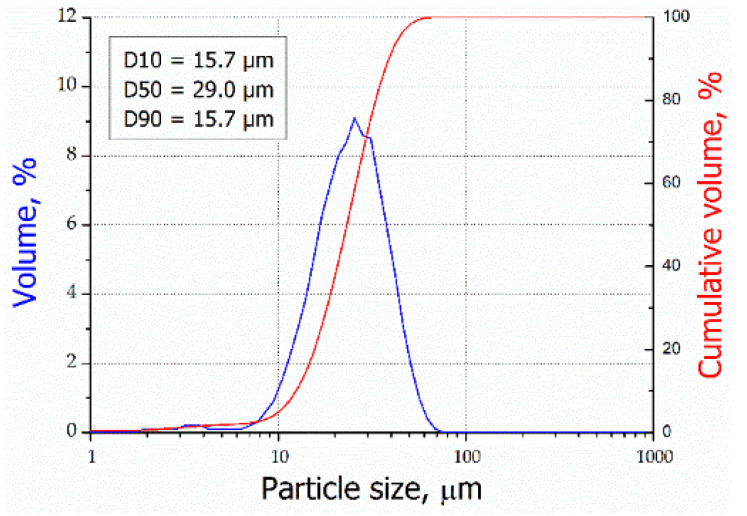
Particle size distributions of the M789 powder.

**Figure 2 materials-15-01734-f002:**
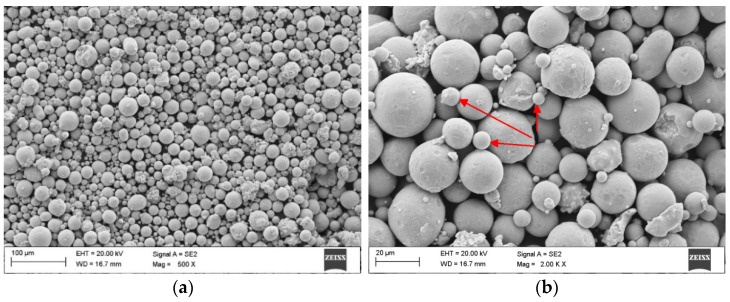
SEM images showing the morphology of the steel powder M789: (**a**) powder particles and (**b**) magnification× 2000, satellites are marked with red arrows.

**Figure 3 materials-15-01734-f003:**
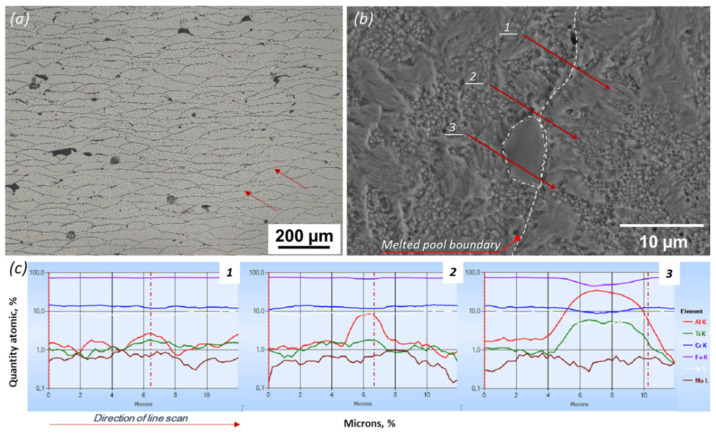
The microstructure of printed M789 in cross-section: (**a**) in as-printed condition (LOM), (**b**) SEM/EDS line analysis on the melted pool boundary (solution annealed condition), (**c**) composition analysis in linear EDS scans along lines 1 ÷ 3 on (**b**), dash-dot line corresponds to the melt pool boundary.

**Figure 4 materials-15-01734-f004:**
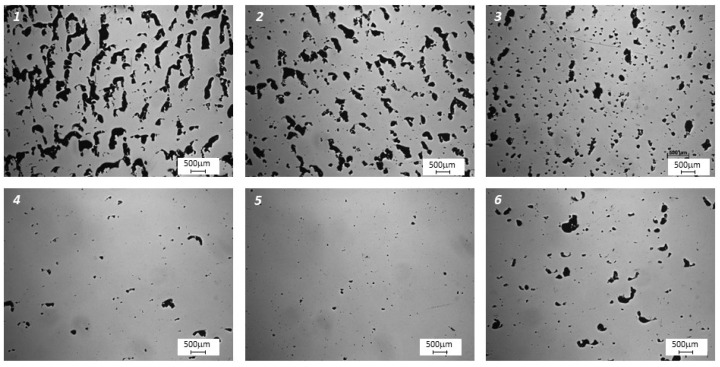
LOM images showing the defects (within the horizontal cross sections) on the nonetched surface of the M789 steel built via AM. Images (**1**) to (**6**) correspond to the sample numbers and printing parameters provided in Table 2.

**Figure 5 materials-15-01734-f005:**
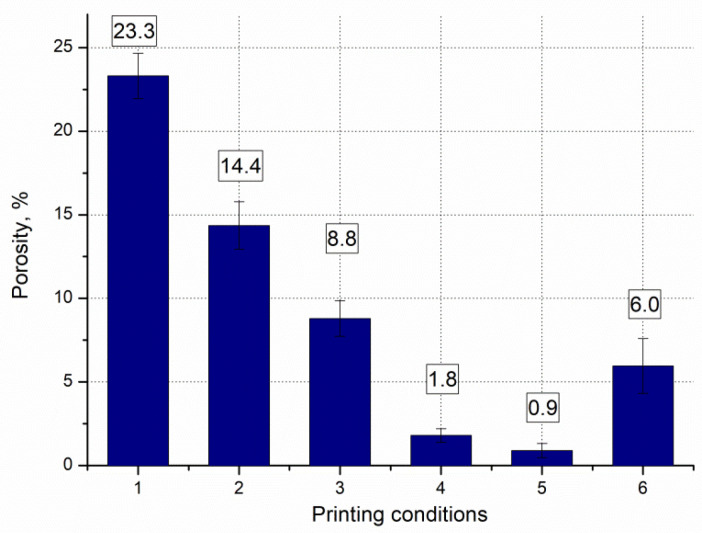
Bar chart showing the influence of the printing parameters (given in Table 2) on the porosity of the M789 steel.

**Figure 6 materials-15-01734-f006:**
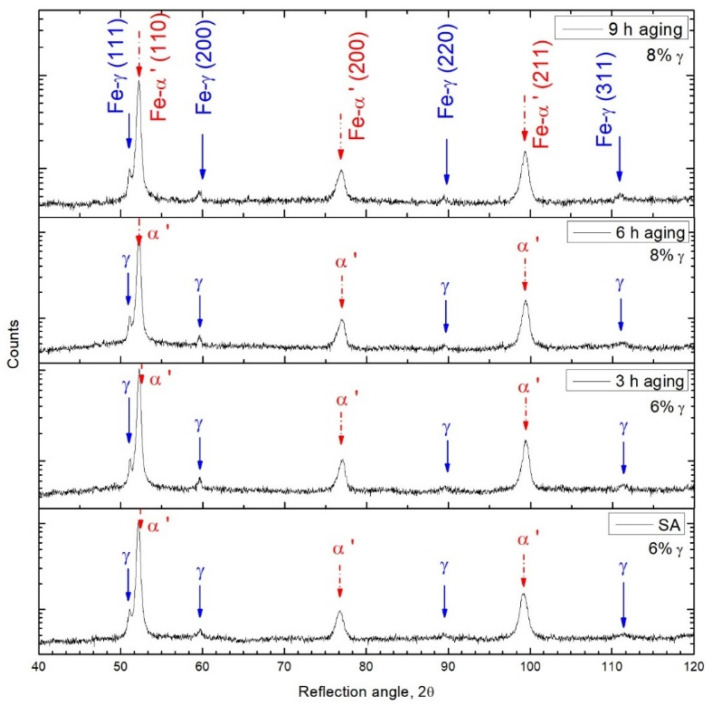
X-ray diffraction patterns for the solution annealed and aged samples.

**Figure 7 materials-15-01734-f007:**
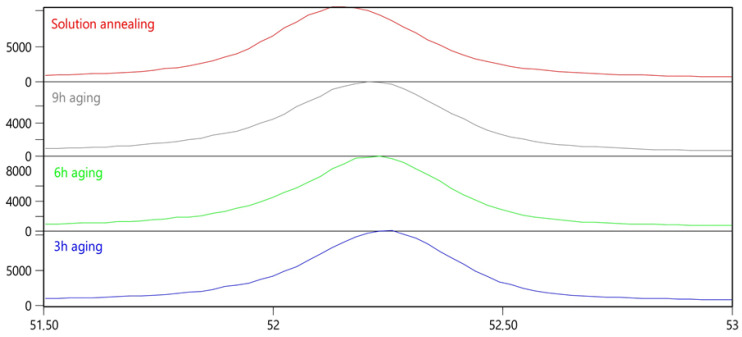
X-ray diffraction patterns near the Fe-α’ (110) peak location.

**Figure 8 materials-15-01734-f008:**
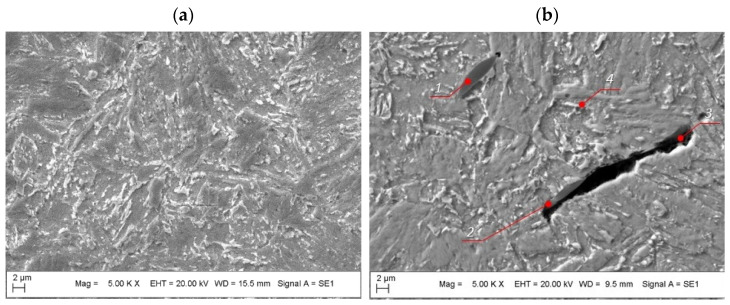
Images of the microstructure of the M789 maraging steel, which are either: (**a**) solution annealed and aged 6 h or (**b**) solution annealed and aged 9 h. Here, at positions 1 and 2, massive precipitates are seen at the grain boundary and secondary porosity, a cavity is found at the border of the melting track at position 3, and the retained austenite (lighter color) is seen between the martensite plates at position 4.

**Figure 9 materials-15-01734-f009:**
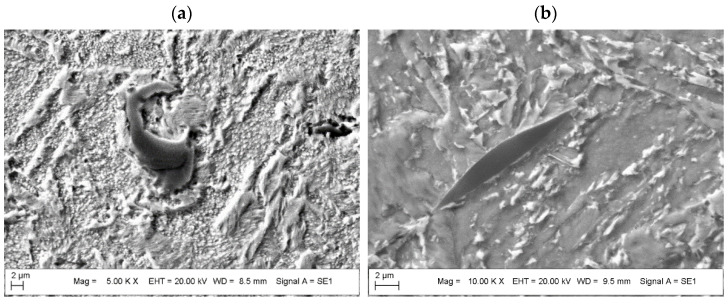
Images of the microstructure of the AM M789 maraging steel with combined oxide inclusions of Ti and Al (TiO_2_:Al_2_O_3_) under: (**a**) solution annealed conditions and (**b**) solution annealed and aged 9 h.

**Figure 10 materials-15-01734-f010:**
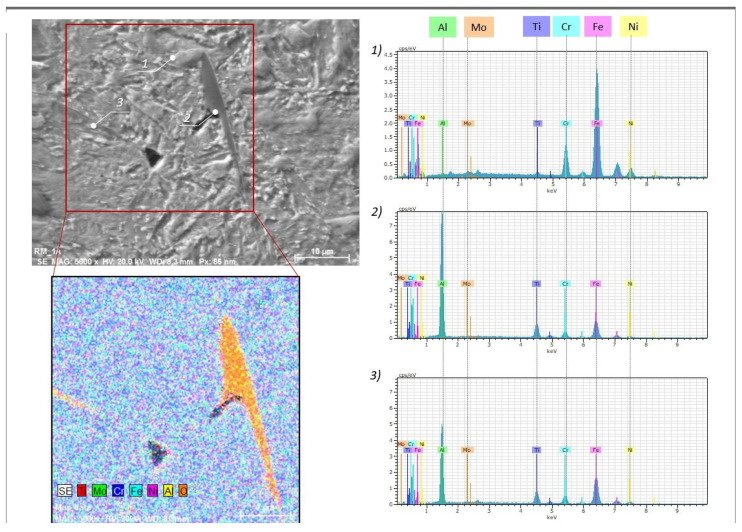
Chemical compositions within the area and from corresponding points marked in white of the M789 maraging steel.

**Figure 11 materials-15-01734-f011:**
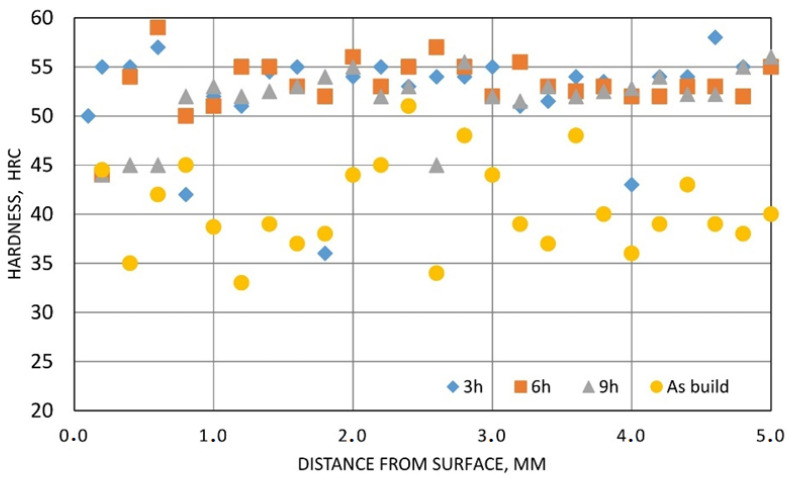
Hardness distribution from surface to core of the AM M789 maraging steel when subjected to aging for 3 h, 6 h or 9 h.

**Figure 12 materials-15-01734-f012:**
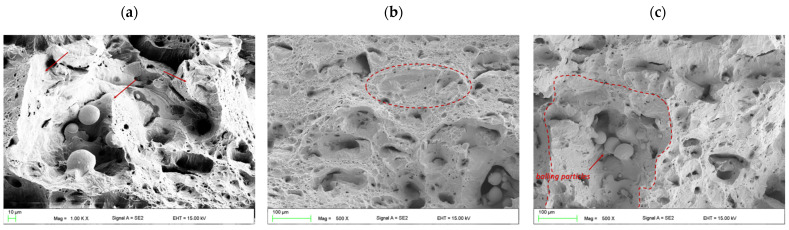
Fractured surface features of the AM maraging steel M789 in solution annealed conditions. Images (**a**–**c**) are described in the text.

**Figure 13 materials-15-01734-f013:**
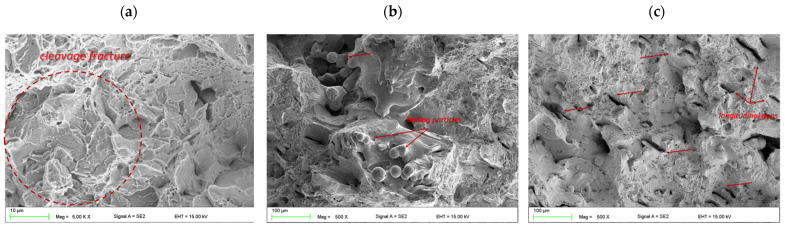
Fractured surface features of the AM maraging steel M789 in aging conditions: (**a**) 3 h, (**b**) 6 h and (**c**) 9 h.

**Table 1 materials-15-01734-t001:** Chemical composition of the M789 maraging steel powder in units of wt% (provided by the supplier).

Element	C	Cr	Ni	Mo	Al	Ti	Co-free
	<0.02	12.2	10.00	1.00	0.60	1.00	

**Table 2 materials-15-01734-t002:** The six sample conditions chosen for the LPBF process, in which the *h*-layer thickness is kept constant at 30 μm.

Sample No.	Laser Speed, mm/s	Laser Power, W	Hatch Distance, mm	Energy Density, J/mm^3^
1	400	155	0.035	370
2	400	155	0.075	170
3	480	200	0.16	68
4	680	200	0.08	126
5	340	200	0.12	163
6	600	180	0.08	125

**Table 3 materials-15-01734-t003:** XRD diffraction pattern parameters.

Condition	Parameter	Fe-α’ (110)	Fe-α’ (200)	Fe-α’ (200)	*D* (nm)	*δ* × 10^−3^ (nm^−2^)	*ε* × 10^−3^	Fe-γ (%)
Solution annealing	2Θ°	52.1212	76.691	99.1116	19.55	3.56	3.77	6
	*β*, Θ°	0.3552	0.903	0.851				
3 h aging	2Θ°	52.19379	76.919	99.3355	16.70	3.84	3.66	6
	*β*, Θ°	0.35532	0.832	0.741				
6 h aging	2Θ°	52.17469	76.861	99.301	15.92	4.02	3.75	8
	*β*, Θ°	0.365	0.803	0.83				
9 h aging	2Θ°	52.162	76.83	99.294	15.51	4.72	4.04	8
	*β*, Θ°	0.3643	0.955	0.762				

**Table 4 materials-15-01734-t004:** The chemical composition of the phases within the microstructure of M789 maraging steel.

Spectrum (Point Analysis in Figure 10)	Chemical Composition, wt%					
	Al	Ti	Cr	Fe	Ni	Mo
Point 1	0.20	0.94	10.44	66.57	8.31	0.56
Point 2	35.87	9.41	5.02	23.14	2.66	0.11
Point 3	24.31	7.77	6.12	32.95	3.81	0.40

**Table 5 materials-15-01734-t005:** Mechanical properties of the printed M789 maraging steel in solution annealed and heat-treated conditions (mean value, standard deviation).

Mechanical Properties	Solution Annealed	Heat-Treated, Aging at 500 °C		
		3 h	6 h	9 h
HRC, superficial	26 ± 4.7	46 ± 3.6	42 ± 2.8	42 ± 3.9
HRC, core,	41 ± 4.5	52 ± 4.9	53 ± 2.7	52 ± 3.3
*R*_p0,2_, MPa	869 ± 8	1607 ± 26	1602 ± 39	1520 ± 22
*R*_m_, MPa	968 ± 11	1610 ± 28	1617 ± 45	1553 ± 34
*A*_5_, %	3.5 ± 0.6	2.1 ± 0.1	2.0 ± 0.2	1.8 ± 0.4
Toughness KV, J	20.6 ± 0.3	10.6 ± 0.02	11.3 ± 0.05	11.4 ± 0.05

## Data Availability

The data presented in this study are available upon request from the corresponding author.

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
