# Peer review of "Microstructural and Mechanical Properties of Novel Co-Free Maraging Steel M789 Prepared by Additive Manufacturing"

_materials, 2022, doi:10.3390/ma15051734_

Round 1

Reviewer 1 Report

  1. Add some literature of other important Steel such as Duplex Stainless steel.
  • EDM performance characteristics and electrochemical corrosion analysis of Co-Cr alloy and duplex stainless steel: A comparative study
  • Examination of hemocompatibility and corrosion resistance of electrical discharge-treated duplex stainless steel (DSS-2205) for biomedical applications.
  1. Figure 5 needs standard deviation results.
  2. Figure 6 should be cleared.
  3. There are so many SEM images. Combined images should be preferred for reader point of view.
  4. Figure numbering is not appropriate, for example- Figure number 8,9,10 is repeated again.

Author Response

The authors want to thank you for the comments and suggestions to the manuscript. We tried to address all your remarks. In attachment, we are sending answers to the comments.

Reviewer 2 Report

The paper focuses on the processing of novel steel by additive manufacturing, being its microstructure and mechanical properties evaluated in terms of some heat treatments. I believe that the results are sound, and can be interesting for the Journal’s readers. Some points must be checked:

  • A minor English language revision is required, once some sentences are not clear enough for a good understanding;
  • Some parts lack references that support the sentences. Some of them are indicated in the annotated manuscript;
  • The study contains hardness measurements to evaluate the mechanical properties of the samples. However, as it is discussed the presence of some intermetallic compounds, the authors are welcome to add micro-hardness values, once it could provide valuable information about the hardness of the solid solution and precipitated phases;
  • In Fig. 3, punctual EDS analysis could provide interesting results about the chemical composition of the melted pools and the grains;
  • 5 is not in the correct size, parts of the axis were cut. It should be revised;
  • Insert the correspondent Greek letters to indicate the phases detected in the XRD patterns. Regarding the intermetallic compounds, why were they not detected by XRD measurement?
  • Table 3: the results could be more exhibited as a graph instead of the table;
  • Table 5: please, add the stress-strain curves of the samples in the paper;
  • Last but not the least, compare the results found with the literature and highlight the importance of the findings for the field. What are the advantages of processed samples for industrial applications?

Taking it into account, I recommend a major revision of the paper.

Author Response

The authors would like to thank you for the comments and suggestions to the manuscript. We tried to address all your remarks. Please find in attachment the answers to the comments.

Reviewer 3 Report

  1. English should be improved. Many sentences that are too long and some ones are too complicated. Confusion with tenses making it difficult to understand the sequence of events. Modal verbs should be used according to their meaning.
  2. When the ‘tensile strength and yield strength’ (Line 30) are mentioned together, it is better to write ‘ultimate tensile strength and yield point’ (in my opinion).
  3. Line 158: What is it ‘powder an as printed samples’?
  4. An introduction part should be added to Section 3.1, which starts too abruptly and is unclear.
  5. Line 169, 174: Used equipment and methodology should be described in the relevant section, but not in the results.
  6. Lines 171-172: 'A chemical analysis EDS confirmed the presence of the main alloying elements in the studied M789 maraging steel.' is a very important scientific achievement? Why are the metal contamination  levels with other elements not studied, which are very important in this case?
  7. In Figure 4, the size markers are too small and illegible.
  8. Lines 189-190: The microstructure of steels, especially in their welds and additively fabricated products, depends on many nuances of their manufacture, including thermal cycling, metal contamination, etc. A claim of microstructure type based on the reference alone is unacceptable in this context, in my opinion, since no data on the chemical composition (including contaminations) of the metal has been reported. The statement ‘an Ar inert gas atmosphere at an oxygen level that is below 10 ppm’ (lines 118-119) in no way guarantees the repeatedly heated metal from contamination with nitrogen, carbon and other elements of the periodic table. This idea was developed by the authors on page 9, but apart from their results for some reasons.
  9. The ‘FWHM’ should be explained at the first mention.
  10. Some misunderstandings with the numbering of figures after the tenth (and the text, as consequence).
  11. The conclusion about the metal contamination effect on its mechanical characteristics is an axiom. For this particular case, at least EDS analysis results should be presented to confirm it. In addition, the influence of hydrogen is also possible (with which the re-heated metal could be saturated) that was completely ignored by the authors.

Author Response

The authors woula like to thank you for the comments and suggestions to the manuscript. We tried to address all your remarks. In attachment we are sending answers to the comments.

Reviewer 4 Report

Authors evaluated microstructural and mechanical properties of Co free steel prepared by additive manufacturing. Following points must be addressed for to further improve the quality of draft. 

  • Figure 5 is misplaced, should be corrected.
  • There is no problem statement in the abstract which led to design this study. 
  • Authors should introduce a summarized table of similar research and their parameters used, along with referred results to further enrich the introduction section. 
  • Authors are recommended to introduce pictorial representation of processing setup. 
  • How authors ensured the variability of results? Add the standard deviation of repetitions. 
  • Tagging the details on the pictures lead readers to follow the discussion. It is recommended. 
  • Authors should recheck the units presented in the discussion. 
  • Introducing future recommendations and limitations of the study are recommended. 

Good luck

Author Response

(The authors gave the same response as above.)

Round 2

Reviewer 1 Report

Consider the manuscript in present form.

Reviewer 2 Report

The authors provided adequate modifications that improved the quality of the paper.

Reviewer 3 Report

The manuscript might be published in the present form.

Reviewer 4 Report

Thanks for improving the manuscript.